# *Aspergillus niger* Fermentation Broth Promotes Maize Germination and Alleviates Low Phosphorus Stress

**DOI:** 10.3390/microorganisms11071737

**Published:** 2023-07-02

**Authors:** Maoxian Tian, Changhui Zhang, Zhi Zhang, Tao Jiang, Xiaolan Hu, Hongbo Qiu, Zhu Li

**Affiliations:** 1College of Agricultural, Guizhou University, Guiyang 550025, China; 17886137478@163.com (M.T.); 18285423484@163.com (C.Z.); zhangzhi11272021@163.com (Z.Z.); jt_2547@126.com (T.J.); 14728622955@163.com (X.H.); 2College of Life Sciences, Guizhou University, Guiyang 550025, China

**Keywords:** *Aspergillus niger* fermentation broth, maize, germination, gene expression, low phosphorus stress

## Abstract

*Aspergillus niger* is a type of soil fungus with the ability to dissolve insoluble phosphate and secrete organic metabolites such as citric acid. However, whether cell-free *Aspergillus niger* fermentation broth (AFB) promotes maize growth and alleviates low-phosphorus stress has not been reported. In this study, we explored their relationship through a hydroponics system. The results indicated that either too low or too high concentrations of AFB may inhibit seed germination potential and germination rate. Under low phosphorus conditions, all physiological indexes (biomass, soluble sugar content, root length, etc.) increased after AFB was applied. A qRT-PCR analysis revealed that the expression of the *EXPB4* and *KRP1* genes, which are involved in root development, was upregulated, while the expression of the *CAT2* and *SOD9* genes, which are keys to the synthesis of antioxidant enzymes, was downregulated. The expression of *LOX3*, a key gene in lipid peroxidation, was down-regulated, consistent with changes in the corresponding enzyme activity. These results indicate that the application of AFB may alleviate the oxidative stress in maize seedlings, reduce the oxidative damage caused by low P stress, and enhance the resistance to low P stress in maize seedlings. In addition, it reveals the potential of *A. niger* to promote growth and provides new avenues for research on beneficial plant-fungal interactions.

## 1. Introduction

Phosphorus is a constituent of a variety of compounds in plants, and is the second most important nutrient required for plant metabolism, growth, and development [1,2,3]. Phosphorus plays a significant role in metabolic processes and coping with environmental stress. Although there are many types of phosphorus-containing minerals, most of them exist in nature in the form of insoluble phosphates, and the availability of soluble phosphorus is limited to meet the needs of plants [4,5]. However, the main limiting factor for more than 40% of global cropland crop productivity is phosphorus deficiency [6,7]. Low-phosphorus stress affects the synthesis of plant carbohydrates, hindering cell division, inhibiting plant growth and development, and strongly reducing plant physiological activity [8,9]. Consequently, since the 1970s, how to cope with reduced crop yields in a low phosphorus environment has become an important research direction [10].

Plant growth-promoting microorganisms (PGPM) include bacteria, fungi, and other microorganisms, which allow plants to improve their own performance and promote nutrient absorption under biotic or abiotic stress conditions, and have potential benefits for the growth and development of seedlings [11,12]. Generally, many phosphate-solubilizing fungi can dissolve insoluble phosphate or promote phosphorus uptake, thereby promoting plant growth [13,14,15]. Among them, Aspergillus, Penicillium, and Trichoderma are the most effective [4,16,17]. Aspergillus species are known as “gifted” microorganisms [18], of which *A. niger* XJ is a filamentous ascomycete widely found in nature [19]. Aspergillus species are widely used for producing functional enzymes (protease, cellulase), active substances (citric acid, chitin), and organic acids [20,21]. Several studies have previously reported that *A. tubingensis*, *A. Niger*, and *A. Niger strains* ITCB-09 exhibited a high capacity to dissolve inorganic phosphates, including through the production of organic acids [4,22]. In recent years, there have been many reports regarding the use of fungal microbial fertilizers; for example, to promote the vegetative growth of sorghum by inoculating soil with *A. niger* through means of bio-carriers [23,24,25,26]. Additionally, there have been reports of microorganisms that utilize fermentation processes to produce organic acids [27]. However, there have been few reports of *Aspergillus niger* fermentation broths for coping with plant-associated biotic or abiotic stress, particularly in maize hydroponic systems.

Maize is an essential food, fodder, and energy crop worldwide. It ranks first among the three main crops in terms of acreage and production. In 2020, the planting area of maize was 41.26 million hectares (National Bureau of Statistics) and plays an important role in the national economy [28]. However, numerous studies have shown that low phosphorus stress significantly reduces maize yield [29,30]. Therefore, in this paper, we use the cell-free fermentation broth of *Aspergillus niger* as a starting point to explore the effects of AFB on seed germination, growth, and development in maize under low phosphorus stress and then examine the physiological and molecular impacts of AFB in alleviating low phosphorus stress in maize seedlings.

## 2. Materials and Methods

### 2.1. Experimental Design

#### 2.1.1. Test Materials

Maize seeds of the J51 inbred line obtained in the spring of 2021 were used for the experiments.

#### 2.1.2. Test Strain

*Aspergillus niger* XJ; (provided by the College of Life Sciences, Guizhou University, Guiyang, China).

#### 2.1.3. Preparation of Fermentation Broth

Potato dextrose agar (PDA) medium: potato 200 g/L, glucose 20 g/L, agar powder 20 g/L, distilled water 1 L, natural pH. Potato dextrose broth (PDB): 200 g of peeled potatoes, 20 g of glucose, and 1000 mL of distilled water; sterilized at 121 °C for 20 min [31].

Strain activation and inoculation: *A. niger* XJ spores were retrieved from glycerol stocks, streaked on PDA solid medium, and then incubated at 28 °C for 5 days. The spores were subcultured twice to activate the *A. niger* XJ strain. One loop was picked with an inoculation loop and inoculated into the PDB medium. The spores were cultured for 5 days, and the mycelia were filtered off to obtain the fermentation broth [32].

#### 2.1.4. Seed Germination Test

The indoor germination test was carried out according to the method described in “Regulations for the Inspection of Crop Seeds” [33]. The seeds with a uniform size, plumpness, and an intact embryo were selected, disinfected with 5% sodium hypochlorite solution for 10 min, rinsed 2–3 times with sterile water, and then evenly arranged with their embryos facing up on a double-layered filter paper in a Petri dish. Each Petri dish containing eight maize seeds was then soaked with AFB prepared in advance with different dilution ratios (Figure 1A). After 24 h, the maize seeds were taken out and placed on sterile Petri dishes containing moist filter papers. The seeds were evenly placed in the Petri dishes with the seed embryos facing up. Seeds were simultaneously soaked in distilled water to serve as the control. The seeds were then incubated in a dark environment in a growth room with a constant temperature and humidity (25 °C, relative humidity of 70%). The seeds in the Petri dish were monitored daily to record the germination and remove the moldy seeds in time. After 3 days and 7 days, the germination rate and germination potential were measured, and the emergence rate was calculated. The root system and shoot length were measured when the plants were at the six-leaf stage. Each treatment was performed with at least three replicates.

#### 2.1.5. Seedling Hydroponic Experiment

Selected maize seeds were surface-sterilized with 5% sodium hypochlorite for 10 min, washed with distilled water, soaked for 8 h, transferred to a germination tray with filter paper, and germinated at 25 °C in an incubator. Ten-day-old seedlings were moved into a self-made hydroponic box for cultivation (the base of the seedlings was wrapped with absorbent cotton, and the upper part of the box was used as a support). Seven seedlings were left in each box and four replicates were set for each treatment. The boxes were first filled with distilled water to adapt the plants to growth for 24 h and then replaced with nutrient solution. Phosphorus-deficient Hoagland nutrient solution was selected for nutrition. The phosphorus source was KH_2_PO_4_. In the low-P treatment, 0.85 mmol/L K_2_SO_4_ was added to supplement the amount of K. For the corresponding treatments, the base of the seedlings was irrigated with AFB (20-fold diluted fermentation broth). A total of 4 treatments were tested (Figure 1B): no AFB + normal P 1 mmol/L (CK); no AFB + low phosphorus 0.001 mmol/L (P); AFB + normal P 1mmol/L (CK+A); AFB + low phosphorus 0.001 mmol/L (P+A) [34].

### 2.2. Determination of Various Physiological and Germination Indices

#### 2.2.1. Determination of Germination Rate and Germination Potential

According to the Inspection Regulations for Crop Seeds [33], after the seeds absorb water and germinate, the germination number and germination energy (Equation (2)) are counted 3 days after sowing, and the final germination number and germination rate (Equation (1)) are calculated 7 days after sowing. The length of the seedling was measured at the six-leaf stage, and then the germination index (GI) (Equation (3)) and vigor index (VI) of the maize seeds were calculated (Equation (4)).
(1)$$Germination\;rate=\frac{number\;of\;germinated\;seeds\;on\;the\;7th\;day}{total\;number\;of\;seeds}\times 100\%$$
(2)$$Germination\;energy=\frac{Number\;of\;germinations\;on\;the\;day\;with\;the\;highest\;germination\;number}{total\;number\;of\;germination\;within\;3\;days}\times 100\%$$
(3)$$Germination\;index\;(GI)=\frac{Number\;of\;germination\;per\;day}{corresponding\;number\;of\;germination\;days}$$
(4)Vigor Index (VI)=Germination index×Fresh seedling weight (or mean seedling length)

#### 2.2.2. Determination of Biomass 

Four plants were randomly selected from each replicate to measure the plant height, stem thickness, longest leaf, fresh weight of each plant, fresh weight of the upper part (leaves and stem), and fresh weight of the lower part (root).

#### 2.2.3. Determination of Root Morphological Index and Related Gene Expression

The root system of the plant was scanned with a scanner (Epson Perfection V850 Pro, Suwa, Japan). The root image analysis software WinRHIZO Tron (Version 2014) was used to calculate the total root length, total root surface area (SurfArea), total projected area (ProjArea), total root volume (RootVolume), root tip number (Tips), and bifurcation number (Forks). Determination of root development gene expression (same as Section 2.2.6).

#### 2.2.4. Determination of Chlorophyll Content

Chlorophyll content was determined using the 95% absolute ethanol extraction method [35].

#### 2.2.5. Determination of the Relative Water Content of Leaves and Soluble Sugar Content

For measuring the relative water content of leaves, a fully expanded leaf was weighed to calculate the initial weight, *W*1. The leaf was then placed in distilled water for 6 h in the dark at 4 °C and its saturated weight, *W*2, was calculated. Following drying, the dry weight, *W*3, was determined and the relative water content was calculated according to the Formula (Equation (5)) [36,37].
(5)Leaf relative water content=(W1−W3) / (W2−W3)×100%

The anthrone colorimetry method was used for the determination of the soluble sugar content [38].

#### 2.2.6. Determination of Antioxidant Activity and Related Gene Expression

Malondialdehyde (MDA) content was determined using the thiobarbituric acid method [39]. SOD content was determined using the NBT photoreduction method; CAT content was determined using the UV absorption method; POD content was determined using the guaiacol method [40]. For the determination of various physiological indexes, leaf samples were collected from maize seedlings at the seven-leaf stage.

##### RNA Extraction and First Strand cDNA Synthesis

RNA was extracted from maize plants at the seven-leaf stage. RNA extraction was performed according to the kit instructions (Omega-Biotek, GA, USA); first strand cDNA synthesis was carried out according to the instructions of HiScript II Q RT SuperMix for qPCR + gDNA wiper (Vazyme, Nanjing, China). 

##### Real-Time Quantitative PCR 

Real-time quantitative PCR was performed on the Bio-Rad platform (CFX96, Hercules, CA, USA) using the Taq pro Universal SYBR qPCR Master Mix kit. The total reaction system was 20 µL: ddH_2_O 8. 2 µL, 2 × Taq pro Universal SYBR qPCR Master Mix 10 µL, upstream and downstream primers (Table 1), each 1µL, and cDNA 1 µL. The PCR parameters were set at the following: 95 °C, 3 s; 95 °C, 3–10 s; 60 °C, 20~30 s; 40 cycles, fluorescence signal acquisition at 72 °C extension phase. The 2^−ΔΔCt^ method was used to calculate the relative expression levels. Each sample was run with three biological replicates and two technical replicates. 

##### Primer Designing

*SOD9*, *CAT2*, *LOX3*, *EXPB4*, and *KRP1* gene sequences were obtained from www.maizegdb.org and www.ncbi.nlm.nih.gov (last accessed on 20 June 2023). The primers were designed using Primer Premier 6 software and the primer sequences were submitted to the NCBI BLAST tool to ensure their specificity. The reference gene GADPH was used for normalization (Table 1).

### 2.3. Data Processing and Analysis

Statistical analysis was performed on the test data using SPSS27 software, and Origin 2022 was used for graphing. The values in the graphs depict mean ± standard deviation.

## 3. Results

### 3.1. Effect of AFB Soaking on Maize Seed Germination Indexes under Different Dilution Ratios

The maize seeds were inoculated with the fermentation broth of *Aspergillus niger* at different dilution ratios. The results indicated that, compared with the control, the germination rate and germination potential of maize seeds under 20-fold dilution increased, but not significantly. However, the germination index significantly increased (*p* < 0.05). There was no significant difference in the vigor index, germination rate, germination index, and vigor index at 20-fold dilution (*p* < 0.05). The germination rate and germination index were ×20 > CK > ×25 > ×5 > ×15 > ×10 (Figure 2A, C); germination potential was ×20 > CK > ×10 > ×5 > ×15 > ×25 (Figure 2B); the vigor index was ×20 > ×25 >×15 > ×10 > ×5 (Figure 2D).

In summary, it can be seen that, compared with the control, AFB treatment at a dilution of 20-fold increased the germination index of maize seeds, but did not reach a significant level for the germination rate, vigor index, and germination potential. The results showed that the application of AFB had the best effect on seed germination at a dilution of 20-fold, thereby indicating that applying a suitable concentration of AFB could promote the germination and emergence of seeds.

### 3.2. The Effect of AFB on the Biomass of Maize Seedlings under Low Phosphorus Stress

Compared with the normal phosphorus conditions, the plant height, stem thickness, upper fresh weight, and fresh weight of maize seedlings under low phosphorus stress (P) were significantly decreased (*p* < 0.05). Under normal phosphorus conditions, the application of AFB significantly increased plant height (*p* < 0.05); however, the increase in stem thickness was not significant. Compared with the low phosphorus stress (P), the plant height and upper fresh weight of the low phosphorus plus fermentation broth (P+A) treatment were statistically insignificant. The stem thickness, the maximum root length, the fresh weight of the lower part, and the fresh weight all significantly increased (*p* < 0.05), which increased by 0.26 cm, 6.74 cm, 0.92 g, and 1.58 g, respectively (Figure 3).

### 3.3. Effects of AFB on Root Morphological Indicators under Low Phosphorus Stress

Compared with the control, the Total Length, the AvgDiam, and the number of root tips per piece of maize seedling under low phosphorus stress were significantly increased (*p* < 0.05); however, the SurfArea, ProjArea, RootVolume, and number of bifurcation Forks increased but were not statistically significant. The Total Length, the AvgDiam, the tip number, the SurfArea, the ProjArea, the RootVolume, and the Forks all significantly increased following the application of AFB under low P stress (P+A) (*p* < 0.05) (Table 2, Figure 4C).

Compared with the control, the expression levels of root-related genes EXPB4 and KRP1 were downregulated under low phosphorus stress (P) (Figure 4A); the expression levels of the two genes were upregulated following the application of AFB under low phosphorus conditions (P+A), especially EXPB4 (Figure 4B).

### 3.4. Effect of AFB on Chlorophyll Content of Maize Seedlings under Low Phosphorus Stress

Compared with normal phosphorus conditions, the total chlorophyll (Chl a+b), chlorophyll a (Chl a), chlorophyll b (Chl b), and carotenoids of maize seedlings under low phosphorus stress were significantly increased (*p* < 0.05). Under low P stress, the application of AFB (P+A) significantly increased the total chlorophyll (Chl a+b), chlorophyll a (Chl a), and carotenoid content of maize seedlings (*p* < 0.05) by 1.17, 0.89, and 0.21 mg/g, respectively. While chlorophyll b (Chl b) had no significant difference (P+A) (Table 3).

### 3.5. Effects of AFB on Relative Water Content, MDA Content, and Soluble Sugar Content in Maize Seedling Leaves under Low Phosphorus Stress

Compared with normal phosphorus conditions, the relative water content of maize seedlings under low phosphorus stress (P) decreased to some extent but not significantly. The application of AFB under low phosphorus stress (P+A) resulted in a significant decrease of 0.05 g in the relative water content of maize seedlings (Figure 5A). Compared with normal phosphorus conditions, the MDA content of maize seedlings under low P stress (P) was significantly increased by 0.0224 µmol/g. However, the application of AFB under low phosphorus stress (P+A) significantly decreased the MDA content by 0.0084 µmol/g when compared with low phosphorus stress conditions (P) (Figure 5B). Compared with the normal phosphorus conditions (CK), the soluble sugar content of maize seedling leaves under low phosphorus stress (P) decreased by around 1.47 mg/g, but the difference was not significant. However, under low phosphorus stress, the application of AFB (P+A) resulted in an increase of 2.45 mg/g of soluble sugar content in seedling leaves when compared with low phosphorus conditions alone (P) (Figure 5C).

### 3.6. Effects of AFB on Antioxidant Enzyme Activity in Leaves of Maize Seedlings under Low Phosphorus Stress

Compared with normal phosphorus conditions, SOD, CAT, and POD activity in maize seedling leaves under low phosphorus stress was significantly increased (*p* < 0.05) by 8.88 U/gFW, 7.88 U/gFW, and 376 U/gFW, respectively. Under low phosphorus stress, the application of AFB (P+A) resulted in significantly reduced (*p* < 0.05) SOD and CAT activity in the leaves of maize seedlings by 8.24 U/gFW and 20.39U/gFW. In contrast, POD activity in the leaves of maize seedlings (P+A) was significantly increased (*p* < 0.05) and increased by 36.3 U/gFW (Figure 6).

### 3.7. Effects of AFB on the Expression of Antioxidant Enzyme Activity-Related Genes in the Leaves of Maize Seedlings under Low Phosphorus Stress

Compared with the control, the expression levels of key antioxidant enzyme genes *SOD9* and *CAT2* were upregulated, while *LOX3,* a gene involved in lipid peroxidation, was downregulated under low phosphorus stress (Figure 7A). The expression of *SOD9* and *CAT2* was downregulated when AFB was applied under low phosphorus conditions (P+A) and the expression of *SOD9* and *LOX3* was downregulated most obviously (Figure 7B).

## 4. Discussion

Seed germination is an important step in the lifecycle of a plant, dictating vigorous growth and successful seedling establishment. Various seed germination-related indices have been employed to study seed germination [41,42]. Among these, the germination rate (GR) reflects the germination number of seeds; the germination energy (GE) reflects the uniformity and speed of seed emergence; the seed vigor index (VI) reflects the quantitative measurement of the overall physiological quality of seeds [43]. Previous studies have found that *A. Niger* fermentation broth and *Bacillus altitudinis* strain WR10 significantly promoted the growth of wheat seedlings and increased the relative germination rate of wheat seeds under salt stress [44,45]. In this study, different concentrations of ABF were used to inoculate maize seeds (Figure 1A), and the results were observed (Figure 2). Different dilution ratios of ABF had different effects on maize seed germination. Under too high or too low dilution of AFB, the germination rate, the germination index, and the vigor index of the maize seeds were reduced. The germination effect of AFB on maize seeds is best when treated with ABF at 20-fold dilution. Sun et al. reported similar results in the study of ryegrass (*Lolium perenne*) [46]. In this study, during the later stages of seed germination, it was observed that seeds exposed to a low dilution ratio exhibited mildew growth. Furthermore, the severity of the mildew increased with lower or higher concentrations. This phenomenon can be attributed to the presence of acidic compounds, such as oxalic acid, tartaric acid, and citric acid [21,47]. These compounds seem to provide ample carbon sources for mold growth. In more severe instances, *Aspergillus niger* reproduces and generates mature spores. Therefore, it can be argued that a suitable concentration of AFB, in which citric acid is contained, has a beneficial effect on plant growth, can enhance the absorption and utilization of phosphorus, and can improve crop resistance and other functions. The specific mechanism needs further study.

In this study, under low phosphorus stress, the stem thickness, maximum root length, fresh weight of the lower part, and fresh weight were significantly increased (*p* < 0.05) when the fermentation broth (P+A) was applied. The biomass accumulation of maize per plant substantially increased after adding different types of low molecular weight organic acids (citric acid, malic acid, oxalic acid) in low phosphorus soil [48]. Moreover, there is evidence that *A. niger* TL-F2 promotes ryegrass biomass and *A. niger* strain XF-1 fermentation broth significantly increases the plant height, fresh weight, and dry weight of *A. fruticosa* [25,49]. At the same time, Nuangmek et al. demonstrated that *T. phayaoense* effectively improves plant development by increasing plant height, as well as shoot and root dry weight values [50]. The results of this study were consistent with those of Si-Yu, who found that inoculating seedlings with phosphorus-solubilizing bacteria significantly increased plant height and dry shoot biomass under low phosphorus stress [51]. In addition, Patil et al. and Rahmansyah et al. also reported similar results [14,52]. 

Low phosphorus stress can significantly disturb the homeostasis of elemental nutrients. At low phosphorus stress, the root system undergoes morphological and physiological adaptations to obtain more phosphorus; e.g., promote primary root elongation and increase the root surface area, the average diameter, and the surface absorption area [53,54]. In this experiment, it was found that the root parameters (total root length, average diameter, number of root tips, total surface area, total projected area, total root volume, and number of bifurcations) were significantly higher after the application of fermentation broth compared with those without fermentation broth (*p* < 0.05) (Table 2; Figure 4). This is consistent with the findings of Yahya et al. [55]. This study is similar to previous studies that reported that *A. niger* inoculation increased the seedling root growth of lettuce, pepper, scarlet eggplant, watermelon, and tomato seedlings [56]. Moreover, the gene expression results in this study show that the expression of *EXPB4* and *KRP1* genes related to root development are upregulated under low phosphorus conditions. In particular, the expression of *EXPB4* was significantly upregulated, suggesting that the application of AFB primed the expression of the *EXPB4* and *KRP1* genes to promote root development and resist low phosphorus stress, and hence, increased various parameters describing root growth. Studies involving the *A. niger* CSR3 strain have indicated that IAA, gibberellin, organic acids, and other metabolites in the fermentation broth play a vital role, capturing more phosphorus and promoting rhizome growth [57]. The study by Mehmood et al. attained similar results [58]. At the same time, the increase in various root parameters after the application of fermentation broth indicates the root system has high efficiency and a larger phosphorus concentration distribution index after the application of AFB [59], thereby alleviating low phosphorus stress.

Carotenoids and anthocyanins are part of the plant’s antioxidant defense system, and an increase in the biosynthesis of these compounds makes plants more resistant to oxidative stress [60]. At the same time, the level of chlorophyll content directly affects the plant leaf’s photosynthesis ability. In this study, the total chlorophyll (Chl a+b), chlorophyll a (Chl a), and carotenoid contents of maize seedlings were significantly increased (*p* < 0.05) after the application of AFB, by 1.17 mg/g, 0.89 mg/g, and 0.21 mg/g, respectively (Table 3). Similarly, an experiment by Lubna et al. found that inoculation of the fungus *A. niger* CSR3 with soluble phosphate significantly increased the chlorophyll content of maize [57]. Meanwhile, it has been demonstrated that inoculation of three newly identified mineral-solubilizing *A. niger* strains increased chlorophyll content in Arabidopsis and onion plants [61]. After inoculation with *Aspergillus flavus*, *A. niger* XF-1 fermentation broth, and irrigation with 100-fold diluted AFB, the chlorophyll content, secondary metabolite content, and carbon assimilation of tomatoes, *Amorpha fruticosa*, and wheat were all found to be enhanced [43,49,62]. Thus, the application of AFB increases the carotenoid and chlorophyll content, thus enhancing the antioxidant capacity of the plant and the photosynthetic capacity of the maize leaves under stress.

Malondialdehyde (MDA) is a byproduct of oxidative damage of membrane lipids in response to reactive oxygen species (ROS) [63]. It can be used as an indicator of biotic or abiotic stress to evaluate the degree of plasma membrane damage and the tolerance of plants to biotic or abiotic stress [64]. This experiment is consistent with previous reports where, under stress, MDA content increased and soluble sugar content decreased [65,66]. Similarly, our results showed that applying AFB under low phosphorus stress reduced MDA content (Figure 5B), which was consistent with the results of Ding et al. [67]. At the same time, the expression of the membrane lipid peroxidation gene *LOX3* was downregulated under this treatment (Figure 7B). This suggests that the fermented broth mitigates the damage to the plant cell membrane caused by the low-phosphorus environment. In addition, we also found that the application of AFB increased the soluble sugar content in maize seedling leaves under low phosphorus stress (Figure 5C). Soluble sugar is important for osmotic adjustment in plants. Water stress, salt stress, cold stress, and other adverse environments significantly affect the soluble sugar content in plants [68,69,70].

Typically, plants are fortified in various ways to cope with biotic and abiotic stresses under different conditions [71], including the deployment of antioxidant enzymes [72], such as ascorbate peroxidase (APX), superoxide dismutase (SOD), catalase (CAT), and peroxidase (POD), thereby playing a crucial role in plant stress tolerance [73]. Therefore, in this experiment, by applying AFB to maize seedlings under low phosphorus stress, the enzymatic activities of SOD and CAT were found to decrease. At the same time, POD activity increased (Figure 6), which is consistent with the results reported by Ding et al. [67]. At the same time, the expression of key genes *SOD9* and *CAT2*, related to the synthesis of antioxidant enzymes, is upregulated under low phosphorus conditions and downregulated after the application of AFB (Figure 7). The gene expression levels were consistent with the results of the quantification of antioxidant enzyme activity. Here, maize seedlings showed high POD activity, indicating that AFB had a minimal effect in reducing POD activity. In addition, SOD is the first line of defense, scavenging reactive oxygen species, and is the most effective. It converts superoxide radicals (O^2-^) into O^2^ and H_2_O_2_ [74,75,76]. Therefore, in this experiment, the SOD and CAT enzyme activities under P+A treatments decreased, indicating that the application of AFB not only activated the antioxidant defense systems, but may also have stimulated other stress-related metabolites [77], which reduced ROS production in plants, thus showing a decrease in related enzyme activities. These results are similar to the findings of Begum et al. [78].

## 5. Conclusions

In conclusion, the germination trend of maize seeds was best at a dilution ratio of 20. Too high or too low a concentration may negatively affect seed germination. The application of *Aspergillus niger* fermentation broth significantly promoted growth, alleviated the oxidative damage to maize caused by low phosphorus stress, maintained normal growth and metabolism, and enhanced the resistance of maize seedlings to low phosphorus stress. In addition, the results revealed the growth-promoting potential of *A*. *niger* and provided a basis for incorporating *A. niger* as an essential component for future bio-fertilizer innovation.

## Figures and Tables

**Figure 1 microorganisms-11-01737-f001:**
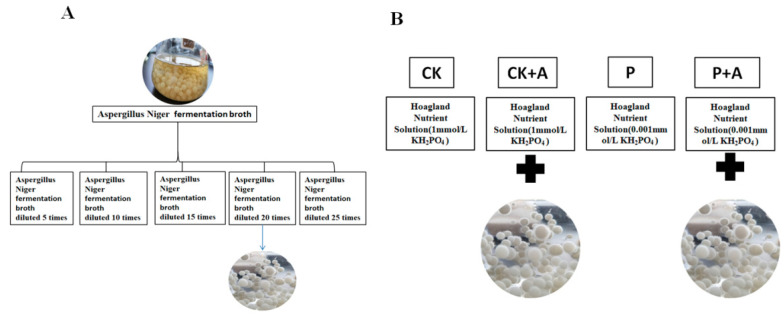
(**A**) The cell-free AFB stock solution was diluted with distilled water 5 times, 10 times, 15 times, 20 times, and 25 times; (**B**) Treatment. CK means no AFB under normal phosphorus conditions (1 mmol/L); CK+A: AFB under normal phosphorus conditions (1 mmol/L); P: no AFB under low phosphorus stress (0.001 mmol/L); P+A: AFB was applied under low phosphorus stress (0.001 mmol/L).

**Figure 2 microorganisms-11-01737-f002:**
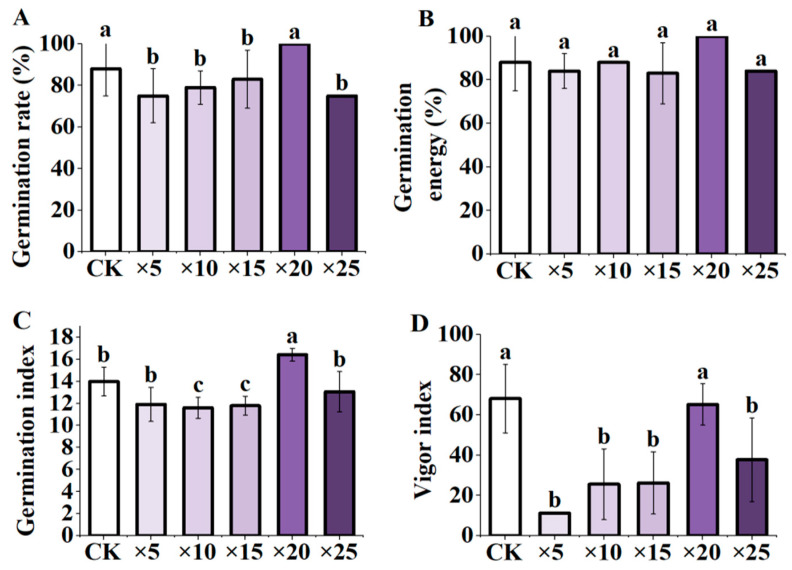
Effects of AFB inoculation on the germination indexes of maize seeds under different dilution ratios. (**A**) germination rate; (**B**) germination energy; (**C**) germination index; (**D**) vigor index; CK: blank control, soaking seeds in distilled water, ×5: 5-fold dilution of fermentation broth, ×10: 10-fold dilution of fermentation broth, ×15: fermentation broth stock 15-fold dilution, ×20: 20-fold dilution of fermentation broth stock solution, ×25: 25-fold dilution of fermentation broth stock solution. Different letters indicate significant differences (*p* < 0.05; one-way ANOVA) and bar values indicate mean ± SD values.

**Figure 3 microorganisms-11-01737-f003:**
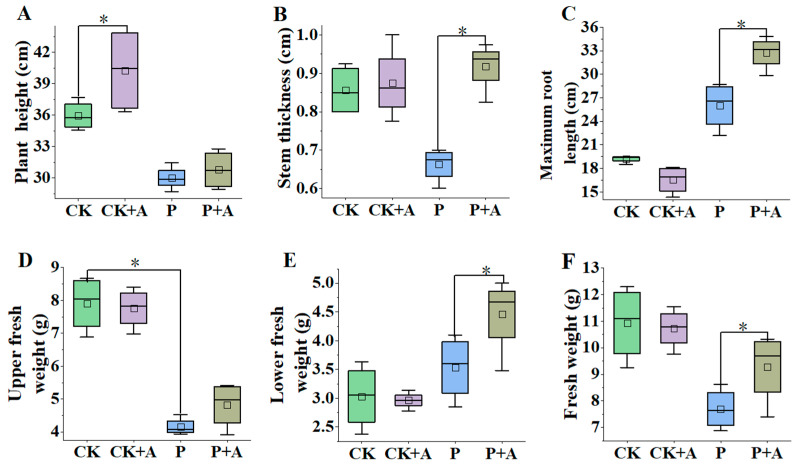
Effects of AFB on plant height (**A**), stem thickness (**B**), maximum root length (**C**), upper fresh weight (**D**), lower fresh weight (**E**), and fresh weight (**F**) of maize seedlings under low phosphorus stress. CK means no AFB under normal phosphorus conditions (1 mmol/L); CK+A: AFB under normal phosphorus conditions (1 mmol/L); P: no AFB under low phosphorus stress (0.001 mmol/L); P+A: AFB was applied under low phosphorus stress (0.001 mmol/L). Asterisks indicate significant differences (*p* < 0.05; one-way ANOVA) and bar values indicate mean ± SD values.

**Figure 4 microorganisms-11-01737-f004:**
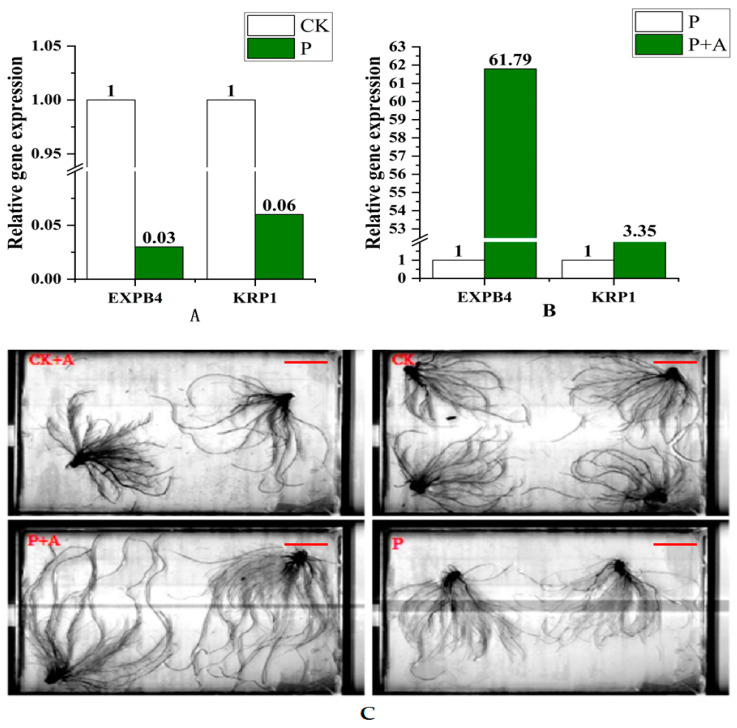
(**A**) The expression levels of EXPB4 and KRP1 genes under low phosphorus conditions. (**B**) Gene expression levels of EXPB4 and KRP1 following application of AFB under low phosphorus conditions; (**C**) Root morphology under different treatments (Scale bars, 5 cm). CK is under normal phosphorus conditions (1 mmol/L) without fermentation broth; P: no fermentation broth was applied under low phosphorus stress (0.001 mmol/L); P+A: Fermentation broth was applied under low phosphorus stress (0.001 mmol/L).

**Figure 5 microorganisms-11-01737-f005:**
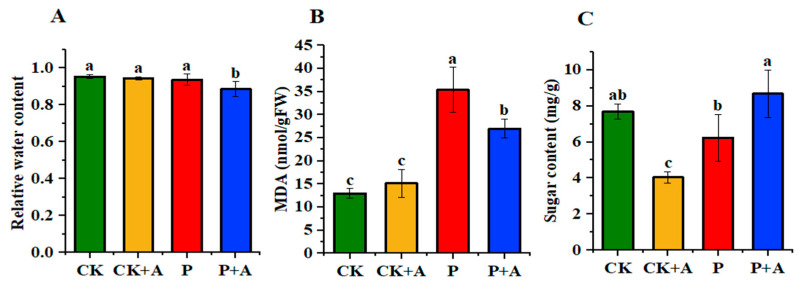
(**A**) Effects of AFB on the relative water content of maize seedlings under low phosphorus stress; (**B**) MDA contents; (**C**) Soluble sugar content. CK means no AFB under normal phosphorus conditions (1 mmol/L); CK+A: AFB under normal phosphorus conditions (1 mmol/L); P: no AFB under low phosphorus stress (0.001 mmol/L); P+A: AFB was applied under low phosphorus stress (0.001 mmol/L). Different letters indicate significant differences (*p* < 0.05; one-way ANOVA) and bar values indicate mean ± SD values.

**Figure 6 microorganisms-11-01737-f006:**
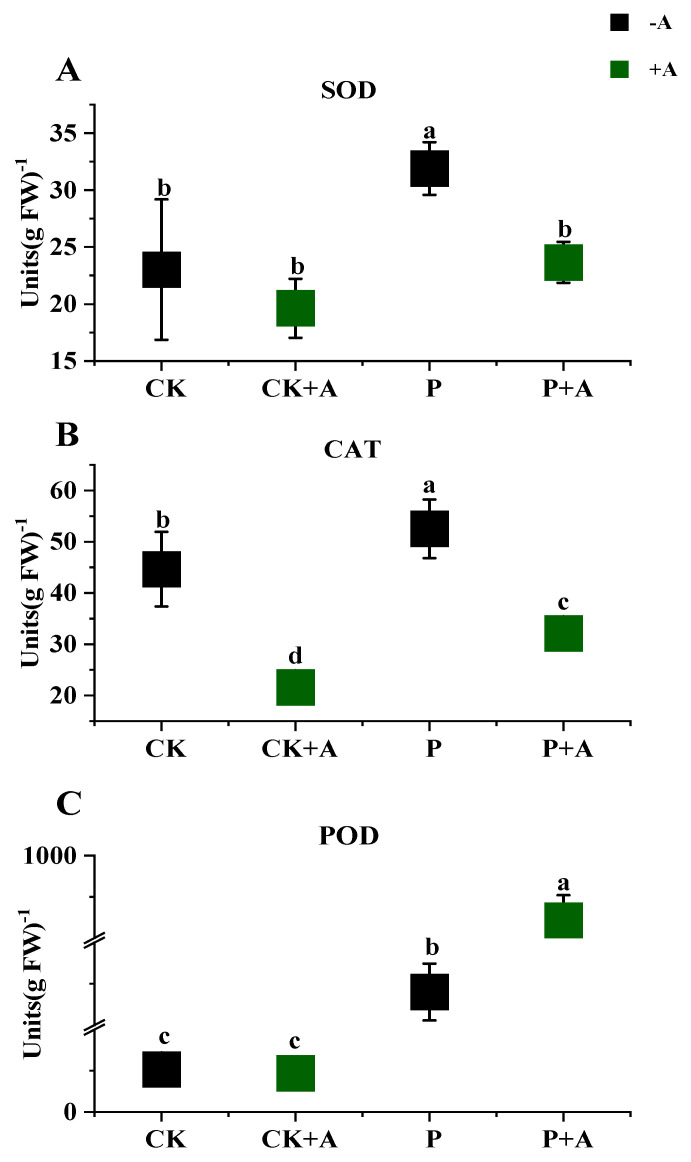
Effects of AFB on antioxidant enzyme activities in the leaves of maize seedlings under low phosphorus conditions. (**A**) SOD activity in maize seedling leaves; (**B**) CAT activity in maize seedling leaves; (**C**) POD activity in maize seedling leaves; CK means no AFB under normal phosphorus conditions (1 mmol/L); CK+A: AFB under normal phosphorus conditions (1 mmol/L); P: no AFB under low phosphorus stress (0.001 mmol/L); P+A: AFB was applied under low phosphorus stress (0.001 mmol/L). Different letters indicate significant differences (*p* < 0.05; one-way ANOVA) and bar values indicate mean ± SD values.

**Figure 7 microorganisms-11-01737-f007:**
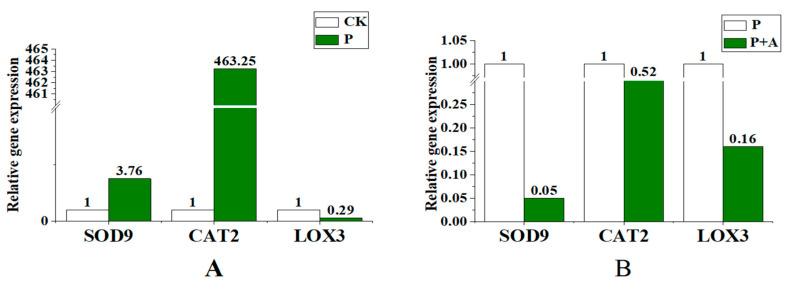
Effects of AFB on the expression of antioxidant enzyme activity-related genes in maize seedling leaves under low phosphorus conditions. (**A**) The expression levels of *SOD9, CAT2*, and *LOX3* under low phosphorus conditions. (**B**) Gene expression levels of *SOD9*, *CAT2*, and *LOX3* following the application of AFB under low phosphorus conditions; CK is under normal phosphorus conditions (1 mmol/L) without fermentation broth; P: no fermentation broth was applied under low phosphorus stress (0.001 mmol/L); P+A: Fermentation broth was applied under low phosphorus stress (0.001 mmol/L).

**Table 1 microorganisms-11-01737-t001:** Primers used for qRT-PCR.

	Forward Primers (5′-3′)	Reverse Primers (3′-5′)
SOD9	GGCTGTTGCTGTGCTTGGTA	CTTGCTCGCAGGATTGTAGTG
CAT2	CCCCAACTACCTGCTGCTAC	TGGTTATGAACCGCTCTTGC
LOX3	CGGCGTTCAAGAGGTTCAG	TGGTCAGAGGTGTTGGGATAGA
EXPB4	CAAGCACACCAACCAGTA	GCACCGAATCTTGTAGCA
KRP1	CCGTATCTCCAGCCATTG	GCCGAGGACCTAGAACAT
GADPH	CCATCACTGCCACACAGAAAAC	AGGAACACGGAAGGACATACCAG

**Table 2 microorganisms-11-01737-t002:** Effects of AFB on the root morphological indexes of maize seedlings under low phosphorus stress.

Treatment	Total Length (cm)	SurfArea (cm^2^)	ProjArea (cm^2^)	AvgDiam (mm)	RootVolume (cm^3^)	Tips	Forks
CK	658.76 ± 151.66 bc	318.53 ± 47.69 b	101.39 ± 15.18 b	1.57 ± 0.18 c	12.38 ± 1.61 b	2874 ± 513 c	6081 ± 2412 b
CK+A	589.14 ± 89.95 c	353.33 ± 67.91 b	112.47 ± 21.62 b	1.87 ± 0.14 c	15.09 ± 1.91 b	1681 ± 402 d	3261 ± 175 c
P	884.05 ± 184.8 b	415.61 ± 43.51 b	132.29 ± 13.85 b	2.93 ± 0.23 b	15.67 ± 0.81 b	6580 ± 556 b	8723 ± 828 b
P+A	1165.63 ± 171.17 a	546.50 ± 92.94 a	173.96 ± 29.58 a	3.64 ± 0.76 a	22.3 ± 3.75 a	9602 ± 1247 a	12,208 ± 1596 a

CK means no AFB under normal phosphorus conditions (1 mmol/L); CK+A: AFB under normal phosphorus conditions (1 mmol/L); P: no AFB under low phosphorus stress (0.001 mmol/L); P+A: AFB was applied under low phosphorus stress (0.001 mmol/L). Different lowercase letters after the data in the same column indicate significant differences among treatments (*p* < 0.05).

**Table 3 microorganisms-11-01737-t003:** Effects of AFB on chlorophyll content of maize seedlings under low phosphorus stress.

Treatment	Total Chlorophyll	Chlorophyll a	Chlorophyll b	Carotenoids
CK	3.33 ± 0.28 c	2.17 ± 0.26 c	1.51 ± 0.09 b	0.17 ± 0.03 c
CK+A	2.72 ± 0.33 c	2.01 ± 0.2 c	1.43 ± 0.08 b	0.18 ± 0.01 c
P	7.82 ± 0.61 b	5.93 ± 0.54 b	3.52 ± 0.49 a	0.45 ± 0.05 b
P+A	8.99 ± 0.57 a	6.82 ± 0.45 a	3.5 ± 0.21 a	0.66 ± 0.07 a

CK means no AFB under normal phosphorus conditions (1 mmol/L); CK+A: AFB under normal phosphorus conditions (1 mmol/L); P: no AFB under low phosphorus stress (0.001 mmol/L); P+A: AFB was applied under low phosphorus stress (0.001 mmol/L). Different lowercase letters after the data in the same column indicate significant differences among treatments (*p* < 0.05).

## Data Availability

Data will be made available on request.

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
