# Peer review of "Aspergillus niger Fermentation Broth Promotes Maize Germination and Alleviates Low Phosphorus Stress"

_microorganisms, 2023, doi:10.3390/microorganisms11071737_

Round 1
Reviewer 1 Report
Some additional, specific comments:
1. What is the main question the study addresses?
Increasing corn yield in conditions of phosphorus deficiency
2. Do you think the topic is original or relevant in the field? Does it address a specific gap in this area?
Due to serious concerns about food security, the study is highly relevant. Originality is obvious and not questioned
3. What does it add to the subject area compared to other published material?
Study of the effect of Aspergillus Niger fermentation broth on corn yield when grown on soils with low phosphorus content. At the same time, the approach under study demonstrates an increase in the seed germination index.
4. What specific improvements should authors consider regarding methodology? What additional controls should be considered?
The methodology and control are absolutely adequate and do not require any modifications.
5. Are the conclusions consistent with the evidence and arguments presented, and do they answer the main question posed?
Conclusions are logical and justified
6. Please include any additional comments on tables and figures.
The quality of tables and figures is at a sufficient level
Некоторые дополнительные, конкретные комментарии:
1. Какой главный вопрос решает исследование?
Повышение урожайности кукурузы в условиях дефицита фосфора
2. Считаете ли вы тему оригинальной или актуальной в данной области? Устраняет ли он конкретный пробел в этой области?
Из-за серьезных опасений по поводу продовольственной безопасности исследование является весьма актуальным. Оригинальность очевидна и не подвергается сомнению
3. Что он добавляет к предметной области по сравнению с другими опубликованными материалами?
Изучение влияния ферментационного отвара Aspergillus Niger на урожайность кукурузы при выращивании на почвах с пониженным содержанием фосфора. В то же время изучаемый подход демонстрирует повышение индекса всхожести семян.
4. Какие конкретные улучшения должны рассмотреть авторы в отношении методологии? Какие дополнительные средства контроля следует рассмотреть?
Методика и контроль абсолютно адекватны и не требуют доработок
5. Соответствуют ли выводы представленным доказательствам и аргументам и отвечают ли они основному поставленному вопросу?
Выводы логичны и обоснованы
6. Пожалуйста, включите любые дополнительные комментарии к таблицам и рисункам.
Качество таблиц и рисунков на достаточном уровне
Author Response
Thank you very much for your review of our article
Reviewer 2 Report
I cannot understand the sense of proposing this paper for publication. It is well-known that soil fungi in general, and Aspergillus niger in particular, may perform phosphorous-solubilizing and other plant growth promoting effects, and that some of their secondary metabolites may rather have a detrimental impact. Indeed, authors demonstrate to be aware of these properties, since they introductorily cite some pertinent literature, including a dedicated review [4]. And what is the point in considering possible application of their strain when it is clearly unable to improve crop performances? Also considering the potential for mycotoxin production by black aspergilli, which can result particularly negative in the case of maize (cfr. 10.3390/toxins2040399).
Moreover, the authors generally denote poor command in carrying out the experimental work. As an example, in the results they repeatedly affirm that the culture broth may induce an increase in several growth parameters, without these effects being statistically significant.
English language and compliance with the MDPI editorial style are also quite approximate, denoting general carelessness.
Author Response
Response :
Thank you very much for reviewing this article. We deeply are indebted for your time and comments regarding our manuscript. For your questions, we have addressed your concerns point by point in the following paragraphs.
I cannot understand the sense of proposing this paper for publication. It is well-known that soil fungi in general, and Aspergillus niger in particular, may perform phosphorous-solubilizing and other plant growth promoting effects, and that some of their secondary metabolites may rather have a detrimental impact. Indeed, authors demonstrate to be aware of these properties, since they introductorily cite some pertinent literature, including a dedicated review [4]. And what is the point in considering possible application of their strain when it is clearly unable to improve crop performances?
Aspergillus niger is a common species of Aspergillus genus, which has rapid multiplication rate and short fermentation cycle. It is one of the safe strains certified by FDA and an important strain for enzyme production. Moreover, Aspergillus niger expression system has excellent protein processing ability, can produce many useful substances such as functional enzymes (protease, cellulase), active compounds, such as citric acid and a variety of proteins and hormones which are directly secreted into the medium.
The main question our research addressed was use of the fermentation broth, not the strain itself. We demonstrated that the secondary metabolites produced by our strain could rescue the seedlings from major stress related symptoms in phosphorus limited conditions. The results from our experiment further demonstrate that the fermented broth could serve as a growth promoting supplement in stressed conditions. And it is also natural that only application of fermented broth cannot compensate for the thousand-fold less phosphorus as phosphorus is one of the building blocks and required for cellular processes. Nevertheless, the fermented broth still significantly improved various plant growth parameters despite a few being non-significant. Therefore, we believe that the research of Aspergillus Niger fermentation broth is of great significance as it improved significantly parameters such as stem thickness, longest root, fresh weight at the bottom, fresh weight and root surface area.
Also considering the potential for mycotoxin production by black aspergilli, which can result particularly negative in the case of maize (cfr. 10.3390/toxins2040399).
First of all, we do not deny the possibility that Aspergillus niger can produce mycotoxin, but the production of mycotoxin varies from strain to strain (Antonia Susca et al., 2016), and no studies have directly demonstrated that Aspergillus niger xj can directly produce mycotoxin. On the contrary, Aspergillus niger has the potential value of producing natural antibacterial active substances
Therefore, we believe that the research of Aspergillus Niger fermentation liquid is of great significance, whether it is for fermentation industry or for producing growth promoting supplements for stressed plants.
Finally, thank you very much for your comments on my article.

Reviewer 3 Report
This study investigates the effect of broth prepared from the fungus Aspergillus niger promoting maize germination in low phosphorus. I have the following comments/queries:
Abstract: niger should have a small n. line 6: explain what is meant by physiological indices. line 7: make it clear whether broth is present or absent as in figure 4A these genes are down regulated in P alone but in P+A they are upregulated, 4B: should not use abbreviations in abstract, line 9, is this malondialdehyde?
Following Keywords add a list of abbreviations.
2.1.4: line 6: do authors mean seeds?
page 3 line 3: how often were there mouldy seeds and how many? I assume the authors started with more than 8, so they can select 8 for their experiments, keeping a constant number. For these experiments treatment was repeated 3 times but for the hydroponic experiments, these were repeated 4 times. Why the difference?
para 2: apart from the first experiments, I assume a broth dilution of 1:20 was always used, make this clear.
Figure 1: the print in this figure needs enlarging. This figure needs a legend, including what the abbreviations stand for.
2.2.1: explain what germination rate, germination potential, germination index and vigour index mean.
2.2.2: why not use all 7 plants?
2.2.5: explain how authors determined soluble sugar content.
page 5: I assume this should be Table 1, not 3.
Figure 2: use either germination potential or energy, legend and B, don't mix them. In legend add concentration of P used.
Page 6 line 7: add control to these?
3.2: Fig 3B and line: use stem thickness or diameter, not both. Again 3C and line 7, use maximum or longest, not both. I find this para confusing, please rephrase and link to figure 3, referring to A-F in text.
page 7 para under Table 2: is this part of the legend for Table 2, if so, add to legend.
Figure 4C add scale and add A, B, C to legend and describe what is in each part.
Page 7: bottom para: line 3 after p<0.05, is this in presence of broth? In text need to make it clear if referring to P or P+A; in some cases both are significantly different from control.
3.4 under Table 3: is this part of legend? In the next para line 6: what do these values refer to? last line, under which condition?
3.5 Table 5: in legend, explain what A,B,C show. Legends should fully describe the table/figure without having to refer to text.
page 9: line 3: is this in presence of broth?
3.6 Figure 6: label each part A,B,C. In legend give SOD, CAT, POD in full and describe each part of figure in full.
3.7: in italics.
Figure 7 legend: line 3: CK+A is not in figure.
para under figure 7 line 5: add in presence of broth.
I have not looked at refs in detail but 32 is incomplete.
In general the quality of the English is good, just a few para that need rewording.
Author Response
This study investigates the effect of broth prepared from the fungus Aspergillus niger promoting maize germination in low phosphorus. I have the following comments/queries:
Response : Thank you for raising these issues. We have also addressed your individual suggestions below. Thank you for your time spent reviewing our work and providing helpful suggestions that have improved the quality of the manuscript.
Point 1 Abstract: niger should have a small n.
Response 1:Thank you ,we have made this correction (Line 1).
Point 2 line 6: explain what is meant by physiological indices.
Response 2:Thank you , The physiological indexes here mainly refer to seedling biomass, root morphology, soluble sugar, etc., which have been modified in the paper(Line 6 and 7)
Point 3 line 7: make it clear whether broth is present or absent as in figure 4A these genes are down regulated in P alone but in P+A they are upregulated, 4B:
Response 3:Thank you for your advice. The results here are all under the condition of Aspergillus Niger fermentation broth(AFB)under low phosphorus. This problem was caused by the incorrect use of punctuation, which has been corrected in the article (Line 6-10).
Point 4 should not use abbreviations in abstract,line 9, is this malondialdehyde?
Response 4: Thank you; Yes, the word “MDA” is malondialdehyde .we have made this correction(Line 10)
Point 5 Following Keywords add a list of abbreviations.
Response 5: Thank you for your suggestion, we have been added after the keyword.
Point 6 2.1.4: line 6: do authors mean seeds?
Response 6: Thank you; yes, that means corn seeds ; we have made this correction (2.1.4: line 6)
Point 7 page 3 line 3: how often were there mouldy seeds and how many? I assume the authors started with more than 8, so they can select 8 for their experiments, keeping a constant number. For these experiments treatment was repeated 3 times but for the hydroponic experiments, these were repeated 4 times. Why the difference?
Response 7: Thank you for your question. In the experiment, seed mold mainly appeared under the treatment of Aspergillus niger fermentation liquid with low dilution ratio, which could be reflected in the germination rate in the article. Three replicates were used to meet the statistical requirements, while the hydroponics test was repeated for four times, mainly considering the sample size required for the subsequent indicator determination, so one more replicate was set.
Point 8 para 2: apart from the first experiments, I assume a broth dilution of 1:20 was always used, make this clear.
Response 8: thank you;this issue was highlighted in 2.1.5: line 12.
Point 9 Figure 1: the print in this figure needs enlarging. This figure needs a legend, including what the abbreviations stand for.
Response 9: thank you, we have added a legend and made corrections (Figure 1).
Point 10 2.2.1: explain what germination rate, germination potential, germination index and vigour index mean.
Response 10: Thank you for your suggestion, we have defined the parameters after 2.2.1.
Point 11 2.2.2: why not use all 7 plants?
Response 11: Thank you for your question, Since the plant morphology and growth were basically the same within the repetition, we believed that the results achieved by selecting 4 and 7 plants were the same. Therefore, 4 plants were mainly recorded as representatives, so as to gain time for follow-up sampling and reduce sample loss.
Point 12 2.2.5: explain how authors determined soluble sugar content
Response 12: Thank you, We that anthrone colorimetric method is used in the material and methods section. Considering that this method would be redundant if introduced in detail, therefore only the reference to detailed methodology is mentioned here (the last line of 2.2.5). Sugar contents through Anthrone colorimetry was determined as follows: Sugar and sulfuric acid form furfural, which further interacts with anthrone to form a blue-green substrate. The color of the substrate is proportional to the sugar content, and the maximum light absorption value was found at 625nm. Therefore, we extracted soluble sugar in maize leaves by anhydrous ethanol, then added anthrone reagent, mixed and boiled in a water bath. Finally, the absorption value at 625nm wavelength was measured by spectrophotometer and the sample sugar content was found by plotting against a standard curve.
Point 13 page 5: I assume this should be Table 1, not 3.
Response 13:Thank you for pointing out the problem there, we have corrected the discrepancy in the article(page 5:table1)
Point 14 Figure 2: use either germination potential or energy, legend and B, don’t mix them. In legend add concentration of P used.
Response 14:Thank you ,we have made this correction(Figure 2)
Point 15 Page 6 line 7: add control to these?
Response 15:thank you, We have made the corrections.
Point 16 3.2: Fig 3B and line: use stem thickness or diameter, not both. Again 3C and line 7, use maximum or longest, not both. I find this para confusing, please rephrase and link to figure 3, referring to A-F in text.
Response 16: Thank you ,we have made this correction(3.2)
Point 17 page 7 para under Table 2: is this part of the legend for Table 2, if so, add to legend.
Response 17: Thank you. We have modified it in this section(page 7 para under Table 2)
Point 18 Figure 4C add scale and add A, B, C to legend and describe what is in each part.
Response 18: Thank you, we have added and modified (Figure 4C)
Point 19 Page 7: bottom para: line 3 after p<0.05, is this in presence of broth? In text need to make it clear if referring to P or P+A; in some cases both are significantly different from control.
Response 19: Thank you for pointing out our question. We have corrected it(the next two paragraphs of the legend in figure 4)
Point 20 3.4 under Table 3: is this part of legend? In the next para line 6: what do these values refer to? last line, under which condition?
Response 20:thank you; I apologize for not expressing its meaning correctly. We have revised it(3.4 under Table 3 and In the next para line 3-6)
Point 21 3.5 Table 5: in legend, explain what A, B, C show. Legends should fully describe the table/figure without having to refer to text.
Response 21: Thank you ,we have made this correction(3.5 figure5)
Point 22 page 9: line 3: is this in presence of broth?
Response 22: Thank you , We have marked in the study(3.5)
Point 23 3.6 Figure 6: label each part A, B, C. In legend give SOD, CAT, POD in full and describe each part of figure in full.
Response 23: Thank you. We have supplemented the picture and modified the legend(3.6 Figure 6)
Point 24 3.7: in italics.
Response 24:Thank you ,we have made this correction(3.7)
Point 25 Figure 7 legend: line 3: CK+A is not in figure.
Response 25:Thank you ,we have made this correction(Figure 7 legend: line 3:)
Point 26 para under figure 7 line 5: add in presence of broth.
Response 26: Thank you, we have been added in this section(para under figure 7 line 5-6)
Point 27 I have not looked at refs in detail but 32 is incomplete.
Response 27:Thank you ,we have made this correction.
